# Endoscopic Anatomy of the Equine Guttural Pouch: An Anatomic Observational Study

**DOI:** 10.3390/vetsci10090542

**Published:** 2023-08-26

**Authors:** Perrine Piat, Jean-Luc Cadoré

**Affiliations:** 1Clinique Equine de Provence, 715 Chemin des Fourches, 13760 Saint Cannat, France; 2Clinéquine, Vetagro Sup, 1 Av. Bourgelat, 69280 Marcy-l’Étoile, France; jean-luc.cadore@vetagro-sup.fr

**Keywords:** guttural pouch, anatomy, endoscopy, equine

## Abstract

**Simple Summary:**

The guttural pouch of the horse is an extension of the auditory tube connecting the throat to the ear. As equine veterinarians, we are actually very fortunate because the guttural pouch provides us with a unique window into numerous important and complex anatomic structures. A thorough knowledge of the underlying structures visible during guttural pouch endoscopy is essential, not only for an accurate diagnosis but also for the treatment of problems in this area. The objectives of this anatomic observational study were to: (1) identify each structure visible on endoscopic examination of the guttural pouch; (2) create a map of the region; and (3) describe a systematic guttural pouch endoscopic examination. Conventional dissections of the guttural pouches of six equine cadavers were first performed, and each adjacent structure was identified. Then, a guttural pouch endoscopic exam of the dissected heads was performed in order to correctly map each visible underlying anatomic structure identified from the previous dissection. Our investigation provides new practical information on guttural pouch endoscopic anatomy and has allowed us to identify structures that were not previously described. We believe that this information may help with an accurate diagnosis in some cases and also potentially reduce iatrogenic trauma to important structures.

**Abstract:**

Guttural pouch endoscopy allows for both the visualization and treatment of many clinical problems in the equine retropharyngeal region. This area is extremely complex, and its description remains a real challenge for veterinary anatomists and practitioners. Six clinically normal equine cadavers were employed. Conventional dissections of guttural pouches without penetrating the guttural pouch membrane were first performed, and each adjacent structure was identified. Then, a guttural pouch endoscopic exam of the dissected heads was performed in order to correctly map each visible underlying anatomic structure identified in the previous dissection. This allowed us to: 1—identify each structure visible on endoscopic examination of the guttural pouch; 2—create a representative map of all the structures accurately identified via endoscopy; and 3—describe a systematic guttural pouch endoscopic examination with practical keys to help identification of the underlying vital structures. Our investigation provides new practical information on guttural pouch endoscopic anatomy and has allowed us to identify structures that were not previously described. We believe that this information may help with an accurate diagnosis in some cases and also potentially reduce iatrogenic trauma to important structures.

## 1. Introduction

The guttural pouch (Diverticulum tubae auditivae) of a horse is a paired extension of the Eustachian tube (Tuba auditiva),, connecting the nasopharynx to the middle ear [1].

As equine veterinarians, we are actually very fortunate because the guttural pouch (GP) provides us with a unique window into numerous important and complex anatomic structures [2] that are impossible to visualize in other species. GP endoscopy allows for the visualization, diagnosis [3,4,5], and treatment [6,7,8,9,10,11,12,13] of many clinical problems in the equine retropharyngeal region.

This area is extremely complex, and its description remains a real challenge for veterinary anatomists [14]. More importantly, equine practitioners and surgeons often have poor knowledge of the endoscopic anatomy of GP structures, despite their importance.

A thorough knowledge of the underlying structures visible during GP endoscopy is essential, not only for an accurate diagnosis but also for the treatment of problems in this area and the limitation of the risk of life-threatening complications [15].

The objectives of this anatomic observational study were to: 

1—Identify each structure visible on an endoscopic exam of the guttural pouch;

2—Create an exact map of the region;

3—Describe a systematic GP endoscopic examination with practical keys to help identify the underlying vital structures.

## 2. Methods

For this purpose, we performed a study in two arms: 

First, we studied the classic anatomy of the GP from the outside in. 

We performed conventional dissections of guttural pouches on 6 clinically normal horses postmortem. The head was disarticulated at the level of the atlanto-axial joint, and the guttural pouches were exposed via a lateral and caudal approach. The guttural pouch membrane was not penetrated. The identity of all the important structures adjacent to each guttural pouch was confirmed by an experienced veterinary anatomist, Pr. Barone.

Then, we studied the endoscopic anatomy from the inside out. 

We performed a guttural pouch endoscopic exam of the dissected head to correctly map each visible underlying anatomic structure identified in the previous dissection. Each structure was visualized and identified from the inside of the pouch via endoscopy, and then its identity was confirmed from the exterior of the pouch by palpation of the different dissected organs. 

This allowed us to create a representative map of all the structures accurately identified via endoscopy.

## 3. Results

Each GP pouch observed endoscopically was different; the thickness of the membrane and the presence of underlying fat made each pouch unique. None of them could show all the underlying anatomic structures.

Thus, we had to pool the different information from each GP observation into a complete representative map that highlights all the identified visible structures (Figure 1). 

We will define below the position and aspect of each underlying anatomic structure by proposing and describing a complete systematic GP endoscopic examination:

A right guttural pouch will be used for the description (all figures).

We suggest first examining the medial compartment and performing a stepwise inspection of four areas: first, the ceiling of the pouch; then, the caudal wall; next the medial wall; and finally, continuing down to investigate the floor of the pouch. We also suggest leaving the nerve assessment until the end, as the anatomy is more complex and variable. And, finally, we will look at the lateral compartment.

Roof of the medial compartment (Figure 2).

−The most clinically relevant structure in this area is the temporohyoid joint, which is a symphysis and links the stylohyoid bone and the styloid process of the temporal bone.−Just close to them lies the petrous part of the temporal bone, which we can see as a large white relief medially to the joint.−Just above the joint, the tensor veli palatini muscle is visible and runs toward the rostral part of the roof of the guttural pouch. −Medially, the internal carotid artery is visible. It forms a sigmoid inflexion before leaving contact with the pouch to join the Circle of Willis.

Caudal wall of the medial compartment (Figure 3 and Figure 4).

Numerous muscles are visible through the transparent membrane:−The occipito-hyoïdeus muscle links the jugular process of the occipital bone to the stylohyoïd bone.−The caudal part of the Digastricus muscle is also inserted into the jugular process but runs ventrally and quickly loses contact with the pouch.−Finally, the stylohyoideus muscle is sometimes visible and runs along the caudal aspect of the stylohyoid bone.−The atlanto-occipital joint lies close to this mass of muscles. It looks like a large circular white surface, more or less concave. Its location is important, as fungal arthritis of this joint has been reported following guttural pouch mycosis [3].

Medial wall of the medial compartment (Figure 5).

It consists of three structures:−The median septum, created by the apposition of the mucous membranes of the two pouches:−The longus capitis muscle caudally, which is a large fleshy muscular body covered by a white and shiny vertical band;−The dorsal pharyngeal recess rostro-ventrally, which is not easily visible from the pouch, but it is important to know its position, especially when performing laser surgery.

Then, we go down to the floor of the pouch.

Floor of the medial compartment (Figure 5)

−The floor lies on the Carotid trifurcation, but this major branching is never clearly visible.−Laterally, we can see the origin of the external carotid artery, which runs under the stylohyoid bone and then goes into the lateral compartment.−Occasionally, it is also possible to visualize the origin of the linguofacial trunk, running cranially, and of the occipital artery, running proximally, between the two carotids.−The stylopharyngeus muscle is particularly evident on the medial and distal aspects of the stylohyoid bone (Figure 4).−The medial retropharyngeal lymph nodes, which can cause GP empyema, lie ventrally, under the floor of the guttural pouch.−Sometimes (as in Figure 4), it is also possible to see the lateral retropharyngeal LN, just close to the internal carotid.

We will now move on to the identification of the different cranial nerves in the medial compartment.

Cranial nerve assessment in the medial compartment (Figure 6).

Their pathways are variable, which can sometimes make their recognition difficult. That is why some points of reference can be useful:

The cranial nerves IX, X, XI, and XII emerge from the cranium behind the ICA and are divided into two pairs of nerves: nerves IX and XII cross the caudal wall obliquely toward the stylohyoid bone, while nerves X and XI course along the ICA.

−The glossopharyngeal nerve is the most dorsal and lateral and is often thinner than its neighbor, number XII. It goes halfway through a ramification that goes down toward the carotid trifurcation: the Hering nerve, or branch of glossopharyngeal nerve, to the carotid sinus, which plays a role in the regulation of blood pressure.

We found that the pharyngeal branch of the glossopharyngeal nerve emerges more distally, under the floor of the pouch, and so is not in contact with the pouch and not endoscopically visible.

−The hypoglossal nerve also runs obliquely, either alone under the glossopharyngeal nerve or joined onto it (Figure 4), in the same mucosal fold. A good way to differentiate them is that distally, the hypoglossal nerve runs behind the external carotid artery, and the glossopharyngeal runs in front of it. Moreover, the hypoglossal nerve is thicker and has no ramifications.−The vagus and accessory nerves are difficult to distinguish as both of them run along or behind the internal carotid artery. A simple way to identify the vagus nerve is to identify its pharyngeal branch, which is easily visible, and go back up to its origin. This branch is the little nerve that we always see running ventrally across the longus capitis muscle, with a rostral direction toward the pharyngeal plexus. Generally, the vagus nerve is lateral to the ICA, and the accessory nerve is medial. 

It is important to know that the vagus nerve also has another branch, the laryngeal branch, which is rarely visible, emerges higher than the pharyngeal branch, and runs distally toward the carotid trifurcation. 

The last part of the examination is the assessment of structures in the lateral compartment.

Lateral compartment of guttural pouch (Figure 7 and Figure 8).

The vessels are the largest structures here:−The external carotid artery (ECA) emerges from behind the stylohyoid bone and gives rise to the others.−The first branch is the caudal auricular artery, which runs up along the stylohyoid bone.−The next branch is the superficial temporal artery.−Following this, the ECA becomes the maxillary artery, which curves and loses contact with the pouch by entering the alar foramen.−The spider web that is visible on the inflection of the maxillary artery is the carotid plexus, which innervates the external carotid and maxillary artery.−Rostrally and ventrally, the maxillary vein runs along the internal pterygoideus muscle.−Then, in the back of the compartment, it is easy to see the styloid process of the auricular cartilage. It looks like a matchstick and moves with the ear.−The facial nerve lies ventrally to this cartilage. It consists of a large horizontal white band that crosses the compartment between the two arteries. It is the largest nerve visible in the guttural pouch. −Just in the triangle formed by the facial nerve and the two arteries, we observe some pink globular structures, which are part of the parotid gland.−Very rarely, it is possible to see the chorda tympani and the maxillary nerve, as in Figure 8.

## 4. Discussion

As reviewed by Freeman [15], complications from the surgical treatment of GP diseases are common and often life-threatening. There is little forgiveness for surgical errors, and so the surgeon should focus on ways to anticipate and prevent them, especially by knowing precisely the anatomy of the underlying structures.

Our investigation provided new practical information on guttural pouch endoscopic anatomy and allowed us to identify structures that were not previously described via endoscopy: the facial nerve, Hering nerve, chorda tympani, mandibular nerve, atlanto-occipital joint, jugular process of the occipital bone, auricular cartilage, and lateral retropharyngeal lymph nodes.

We also tried to describe a practical way to localize and recognize the complex nervous and arterial networks underlying the GP in order to help each veterinarian or surgeon during his/her daily work.

However, there are several limitations to our study. First, the dissections were performed on only six normal cadavers. Therefore, this study is only observational, and no statistics on the anatomic variability of the GP were produced. This could be the subject of a further prospective in vivo study. Second, the assessment was performed on normal, healthy heads; inflammation and different pathologies could interfere with or modify our normal findings. Finally, the dissection and identification of each underlying anatomic structure were performed by an anatomist and are dependent on subjectivity and human error. However, we significantly reduced this risk by having one of the most renowned and experienced anatomists in the world perform the dissections: Pr. Robert Barone.

To conclude, we believe that this information may help with an accurate diagnosis with a complete lesional assessment in some cases. It can also potentially reduce iatrogenic trauma to important structures during surgeries, whether through transendoscopic techniques [6,7,10,12,13] or conventional open approaches [4,11].

## Figures and Tables

**Figure 1 vetsci-10-00542-f001:**
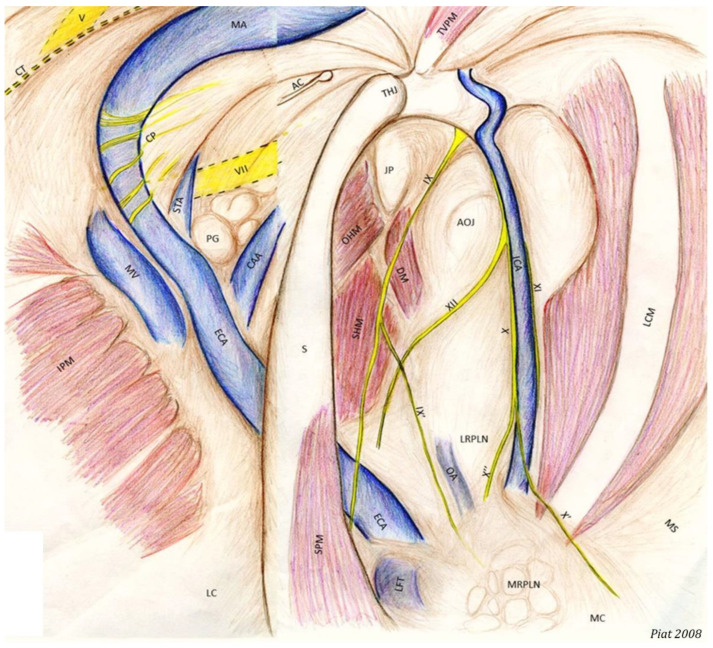
Representative map of all the anatomic structures visible from the right guttural pouch (panoramic view) (ECA: External carotid a.; ICA: Internal carotid a.; OA: Occipital a.; LFT: Linguofacial trunk; CAA: Caudal auricular a.; STA: Superficial temporal a.; MA: Maxillary a.; MV: Maxillary v.; DM: Digastric m.; LCM: Longus capitis m.; OHM: Occipitohyoideus m.; IPM: Internal pterygoideus m.; SHM: Stylohyoideus m.; SPM: Stylopharyngeus m.; TVPM: Tensor veli palatini m.; V: Mandibular n.; VII: Facial n.; CT: Chorda tympani n. (double dashed line, not visible); IX: Glossopharyngeal n.; IX’: Hering n.; X: Vagus n.; X’: Pharyngeal branch of X; X’’: Cranial laryngeal n.; XI: Accessory n.; XII: Hypoglossal n.; CP: Carotid plexus; AOJ: Atlanto-occipital joint; THJ: Temporohyoid joint; JP: Jugular process of the occipital bone; Stylohyoid bone; AC: Auricular cartilage; PG: Parotid gland; MRPLN: Medial retropharyngeal lymph nodes; LRPLN: Lateral retropharyngeal lymph nodes.

**Figure 2 vetsci-10-00542-f002:**
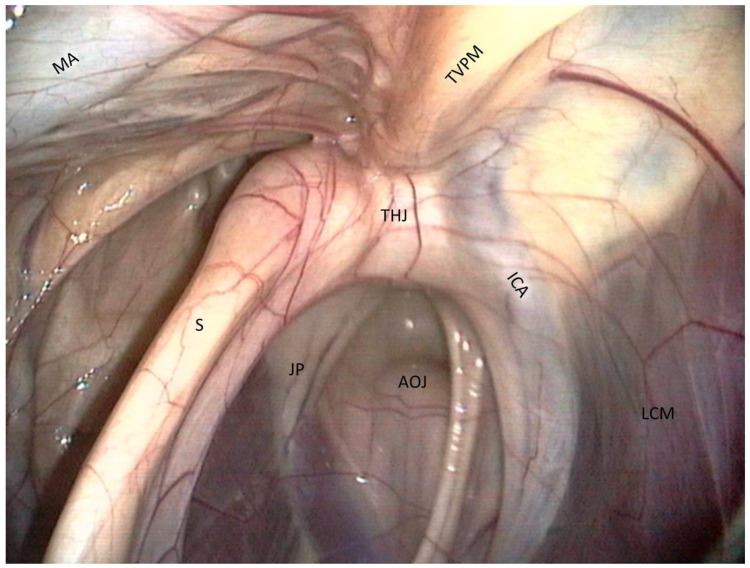
Roof of the medial compartment. (ICA: Internal carotid a.; MA: Maxillary a.; LCM: Longus capitis m.; TVPM: Tensor veli palatini m.; AOJ: Atlanto-occipital joint; THJ: Temporohyoid joint; JP: Jugular process of the occipital bone; S: Stylohyoid bone).

**Figure 3 vetsci-10-00542-f003:**
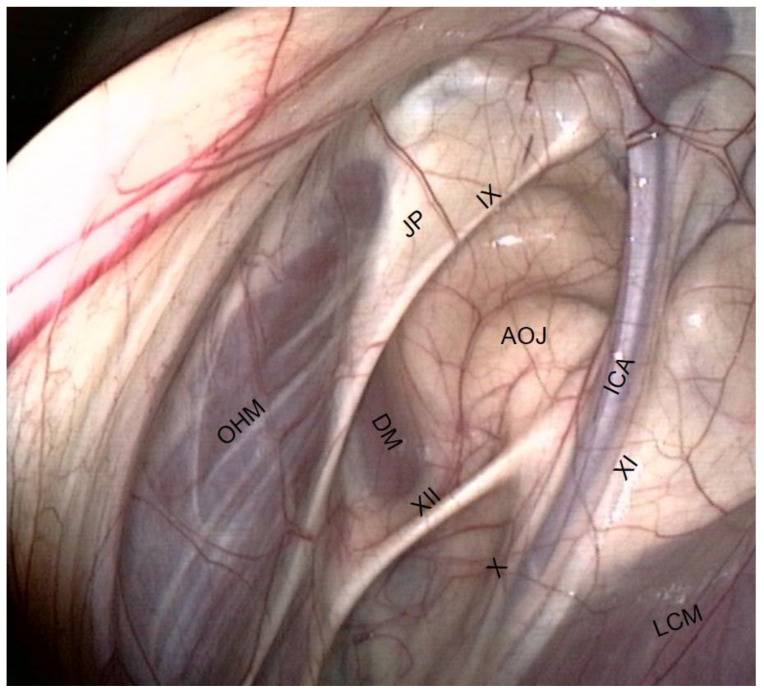
Caudal wall of the medial compartment (dorsal part). (ICA: Internal carotid a.; DM: Digastric m.; LCM: Longus capitis m.; OHM: Occipitohyoideus m.; IX: Glossopharyngeal n.; X: Vagus n.; XI: Accessory n.; XII: Hypoglossal n.; AOJ: Atlanto-occipital joint; JP: Jugular process of the occipital bone).

**Figure 4 vetsci-10-00542-f004:**
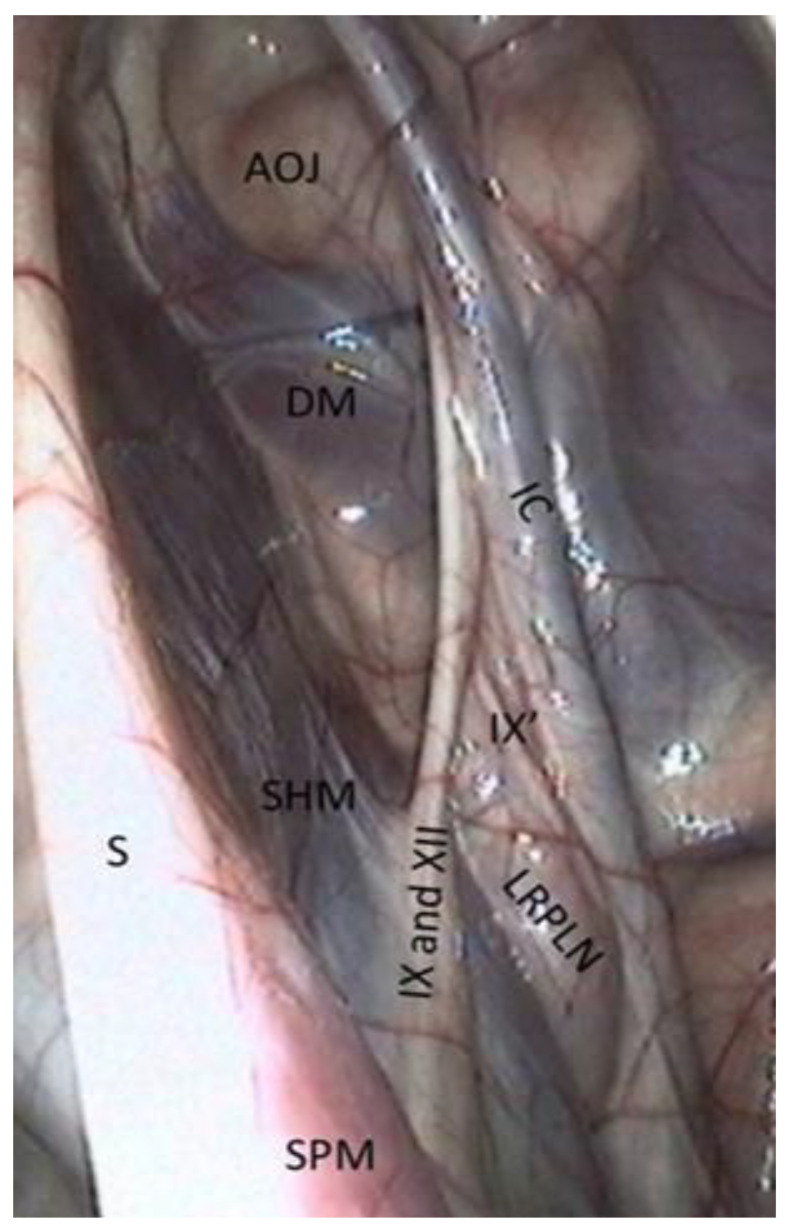
Caudal wall of the medial compartment (lateral part). (IC: Internal carotid a.; DM: Digastric m.; SHM: Stylohyoideus m.; SPM: Stylopharyngeus m.; IX: Glossopharyngeal n.; IX’: Hering n.; XII: Hypoglossal n.; AOJ: Atlanto-occipital joint; S: Stylohyoid bone; LRPLN: Lateral retropharyngeal lymph nodes).

**Figure 5 vetsci-10-00542-f005:**
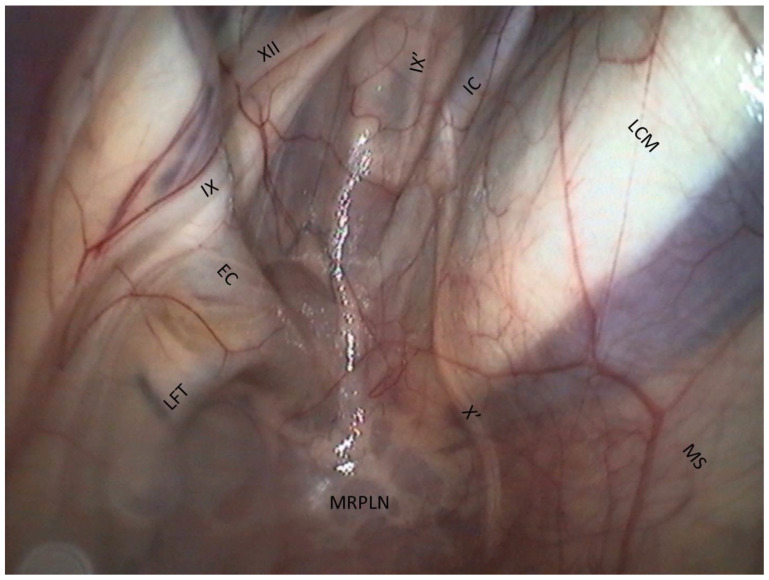
Medial wall and floor of the medial compartment. (EC: External carotid a.; IC: Internal carotid a.; LFT: Linguofacial trunk; LCM: Longus capitis m.; IX: Glossopharyngeal n.; IX’: Hering n.; X’: Pharyngeal branch of X; XII: Hypoglossal n.; MRPLN: Medial retropharyngeal lymph nodes; MS: Median septum).

**Figure 6 vetsci-10-00542-f006:**
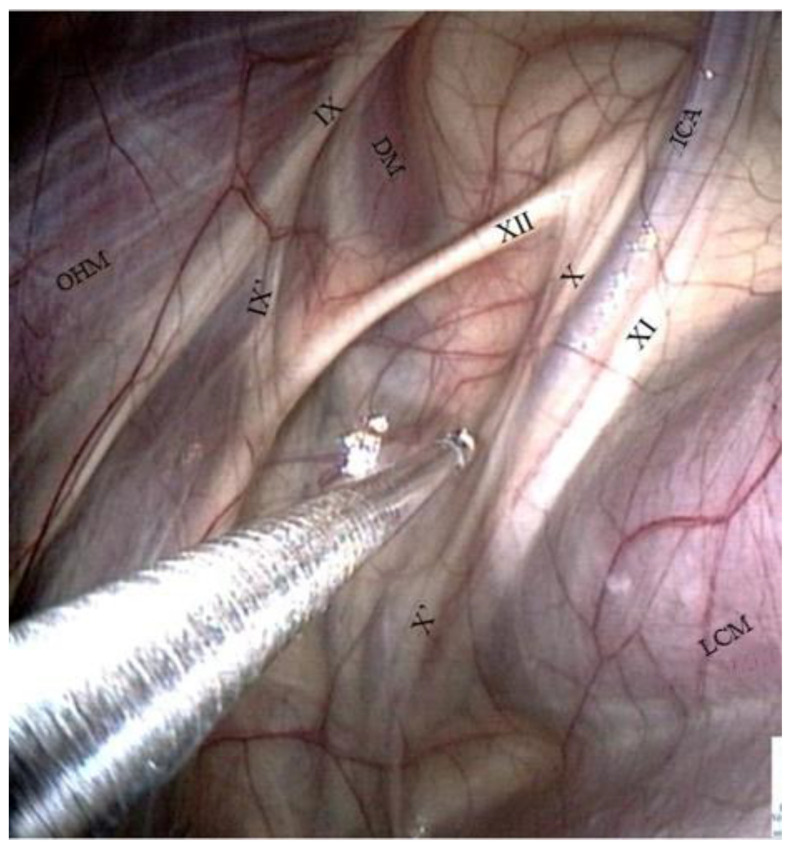
Caudal wall of the medial compartment (ventral part). (ICA: Internal carotid a.; DM: Digastric m.; LCM: Longus capitis m.; OHM: Occipitohyoideus m.; IX: Glossopharyngeal n.; IX’: Hering n.; X: Vagus n.; X’: Pharyngeal branch of X; XI: Accessory n.; XII: Hypoglossal n.).

**Figure 7 vetsci-10-00542-f007:**
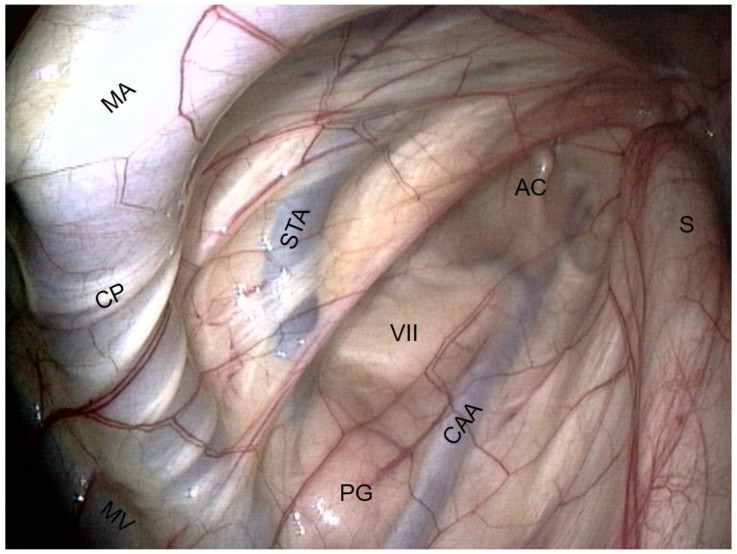
Lateral compartment (large view). (CAA: Caudal auricular a.; STA: Superficial temporal a.; MA: Maxillary a.; MV: Maxillary v.; VII: Facial n.; CP: Carotid plexus; S: Stylohyoid bone; AC: Auricular cartilage; PG: Parotid gland).

**Figure 8 vetsci-10-00542-f008:**
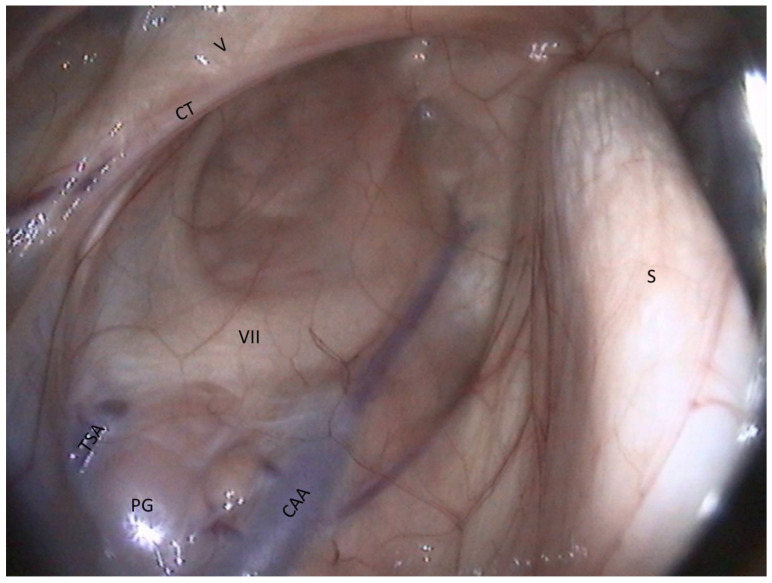
Lateral compartment (dorsal part). (CAA: Caudal auricular a.; TSA: Temporal superficial a.; V: Mandibular n.; VII: Facial n.; CT: Chorda tympani n.; S: Stylohyoid bone; PG: Parotid gland).

## Data Availability

Not applicable.

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
