# Peer review of "Endoscopic Anatomy of the Equine Guttural Pouch: An Anatomic Observational Study"

_vetsci, 2023, doi:10.3390/vetsci10090542_

Round 1
Reviewer 1 Report
Dear authors,
Thank you for this sound presentation of the endoscopic anatomy of the equine guttural pouch. Indeed, it is a complex anatomical structure and very hard to visualize in an anatomic manner.
I only have some minor comments that will allow for an even better comprehension by the readers, along with some linguistic and textual suggestions.
Could you please provide the full terms of the annotations in each figure legends? These are now provided at the end of the manuscript, but this does not allow a smooth reading of the manuscript. So, each abbreviated anatomical term should be followed by the full term in between brackets.
Line 7: "allows both" should be "allows for both"
Line 9: "horses' cadavers" should be "equine cadavers"
Line 21: "guttural" not in bold
Line 24, 25 and other: "auditory tube diverticulum", Eustachian tube, ... Please provide (in between brackets) the official terms that can be found in the Nomina Anatomica Veterinaria.
Figure 1: Is this a caudal or a rostral view? Please mention this. Where is the guttural pouch? Please highlight or delineate.
General remark: Some anatomical term start with a capital letter. Please be consistent.
Line 148: Please put "also" in front of "runs"
Line 167: "vue" should be "view"
Line 198: "cord" should be "chorda"
General: Please remove the double spacings that are here and there present at the start of a new sentence.
Line 203: "his" should be "his/her"
The quality of English is good. I have already made some suggestions above.
Author Response
Thank you for your report
Please see the attachment
Best regards

Reviewer 2 Report
Manuscript review
Title: Endoscopic anatomy of the equine guttural pouch: anatomic observational study.
General comment:
I think this is an impactful manuscript as the information is not well presented in previous manuscripts or textbooks. Because the images are so important since this is an anatomic observational study, most of my comments have to do with the images.
- For all figures, the legends are inappropriate. For the reader, it is painful to have to go back and forth between each picture and the list of abbreviations at the end of the manuscript. I would STRONGLY encourage the authors to list in each figure legend every acronym and to list the meaning for each.
- Some of the abbreviations are listed differently for different images. For example, in Figure 3, the occipitohyoideus muscle is listed with the correct abbreviation as OHM, but in figure 6 it is listed as MOH (there is no MOH listed on the list of abbreviations. Same for the digastricus muscle and others). So please check the abbreviations and use the same consistently throughout the manuscript.
- For each figure legend, I recommend listing all the abbreviations on that figure. For example on Figure 3, there are 10 abbreviations on the figure, but only 4 spelled out and listed in the legend.
- In the legend, do not capitalize the structures. It is fine to bold them, but capitalization is present in some and not others.
- The numeration in the text does not correspond to the figure which is confusing. For example, on p.5 the heading is: 3-Medial wall of medial compartment, but this corresponds to Figure 5. Maybe avoid a numeral listing of the headings?
Comments for individual figures:
- Figure 1: I like the figure but it looks like it was drawn with color pencils and doesn’t look professional. I will leave it to the editor to decide whether it is suitable, but I would recommend using a medical illustrator. Also, on the top left of the figure, there is a double dashed line that is not explained in the legend.
- Figure 2: It would be useful to indicate (perhaps with an arrow) the entrance of the ear canal above the temporohyoid joint, as sometimes exudate can be seen coming from that area.
- Figure 5: The legend says that the stylopharyngeus muscle is really evident, but it is not marked on the figure.
- Figure 6: It would be best if a different picture without the probe could be used, to keep from obscuring structures. This is also the figure where abbreviations are not listed the same way as in the other figures.
Finally, a few typos:
Line 34: Remove “However” and start sentence with “A thorough knowledge…”
Line 128: “Sometimes when they are inflamed (as in Figure 4)- I think the LRPLN in figure 4 is normal, not enlarged, but the way this sentence reads it makes it seem like it is abnormal.
Line 178: spider (not spides)
Line 187: we observe
Overall I really like the paper, and look forward to these minor revisions.
Author Response
Thank you for your report
Please see the attachment,
Best regards
